# New Horizons in Structural Biology of Membrane Proteins: Experimental Evaluation of the Role of Conformational Dynamics and Intrinsic Flexibility

**DOI:** 10.3390/membranes12020227

**Published:** 2022-02-16

**Authors:** Robbins Puthenveetil, Eric T. Christenson, Olga Vinogradova

**Affiliations:** 1Section on Structural and Chemical Biology of Membrane Proteins, Neurosciences and Cellular and Structural Biology Division, Eunice Kennedy Shriver National Institute of Child Health and Human Development, National Institutes of Health, 35A Convent Dr., Bethesda, MD 20892, USA; 2OriGene Technologies, Inc., Rockville, MD 20850, USA; echristenson@origene.com; 3Department of Pharmaceutical Sciences, School of Pharmacy, University of Connecticut, Storrs, CT 06269, USA

**Keywords:** membrane proteins, membrane protein structure, membrane protein dynamics, NMR, cryo-EM, X-ray, FRET, EPR, HDX-MS, antibody fragments

## Abstract

A plethora of membrane proteins are found along the cell surface and on the convoluted labyrinth of membranes surrounding organelles. Since the advent of various structural biology techniques, a sub-population of these proteins has become accessible to investigation at near-atomic resolutions. The predominant bona fide methods for structure solution, X-ray crystallography and cryo-EM, provide high resolution in three-dimensional space at the cost of neglecting protein motions through time. Though structures provide various rigid snapshots, only an amorphous mechanistic understanding can be inferred from interpolations between these different static states. In this review, we discuss various techniques that have been utilized in observing dynamic conformational intermediaries that remain elusive from rigid structures. More specifically we discuss the application of structural techniques such as NMR, cryo-EM and X-ray crystallography in studying protein dynamics along with complementation by conformational trapping by specific binders such as antibodies. We finally showcase the strength of various biophysical techniques including FRET, EPR and computational approaches using a multitude of succinct examples from GPCRs, transporters and ion channels.

## 1. Introduction

A third of both pro- and eukaryotic proteomes consist of membrane proteins. Housed in a milieu of hydrophobic molecules, they serve as crucial contacts of communication between the cytoplasm and non-cytosolic environments, making them essential pharmaceutical targets. While membrane proteins are notoriously difficult to investigate at any level, high-resolution structures of these targets only became feasible at the very end of the twentieth century. It was not until robust technological developments in the fields of X-ray crystallography, NMR spectroscopy and cryo-EM, that the scientific community at large, finally gained access to an ever-increasing number of atomic resolution structures, and began to rationalize how membrane proteins accommodate their function. As if the lack of structural information wasn’t enough to hamper progress, a higher level of complexity arose from the modern understanding of “one structure—one function” paradigm, a primitive simplification useful at the dawn of the scientific era, that has promptly lost credence to the complex maneuvers of membrane proteins. Proteins in general are not static and some even require considerable flexibility to perform their functions. Integral membrane proteins (IMPs), particularly involved in transport and signal transduction, are highly dynamic, lipid-embedded entities that serve as molecular nano-machines controlling bi-directional transport of materials and information across the cell membrane. These physicochemical constraints necessitate a set of well-defined and specific movements, where the interplay between conformational dynamics and intrinsic flexibility becomes extremely important in defining their functionality. The inherent dynamic nature of cell surface receptors, ion channels and transporters are reflected by their unique and distinguishable energy landscape profiles. From a theoretical perspective supported by experimental data for ion channels [1], or G-protein coupled receptors (GPCRs) [2], we can envision a combination of conformational states that remain in equilibrium. Typically, a membrane protein fluctuates between closed and open states, interspersed with transiently active or inactive intermediates. In addition, an ensemble of sub-states constitutes each of these major states, with the sub-states themselves potentially undergoing thermal fluctuations. Thus, it is imperative to utilize all available biophysical approaches to define the details of this hypothetical scheme. Despite spectacular technological advancements in recent years, a thorough quantitative understanding of IMP dynamics remains elusive. In this review, we present the remarkable undertakings by the scientific community integrating multidisciplinary efforts that elucidate the relationship between structure, dynamics, and function using GPCRs, ion channels and transporters as examples.

## 2. Nuclear Magnetic Resonance

Solution NMR (nuclear magnetic resonance) is one of the high-resolution techniques which allows for the investigation of both structural and dynamic properties of biological macromolecules. Being a solution technique, in contrast to X-ray crystallography or cryo-EM, NMR can be performed in environments that closely resemble native cellular conditions. Studying GPCRs, ion channels or transporters, etc., even in the best reconstitution media (reviewed in [3]) with the most advanced TROSY (transverse relaxation-optimized spectroscopy)-based methods [4], still requires perdeuteration and often does not provide desirable resolution and/or sensitivity. Fortunately, with the growing number of crystal/cryo-EM structures in the Protein Data Bank (PDB), we now have a detailed perspective on certain conformers, correlated with specific binding modes or activation states, for many intriguing targets. These atomic resolution structures offer the luxury to focus on specific sites within the protein most sensitive to conformational changes using molecular labels. Although this could be achieved by ^1^H-^13^C methyl TROSY analysis (reviewed in [5]) in deuterated samples, the easiest way is to employ ^19^F NMR.

Several favorable nuclear spin properties of fluorine-19 (^19^F), that have been exploited since the early days [6] are finding stronger footholds again. These include:High magnetic sensitivity: the ^19^F isotope, in contrast to proton (^1^H), the gold standard, has a high relative sensitivity of ~83% and a 100% natural abundance (note that relative sensitivity for detection in NMR experiments at a constant number of nuclei is roughly proportional to the cube of their gyromagnetic ratios [7], (γ_F__19/_γ_H1_)^3^ = 0.94^3^ = 0.83).The lack of endogenous fluorine and thereby the absence of a background signal.Sensitivity to local environment: ^19^F exhibits large chemical shifts dispersion (CSD) which spans 2000 ppm in comparison to a meager 13 ppm for ^1^H; although chemical shifts, arising solely from local van der Waals electrostatic and solvent interactions, typically vary between 2.5 (CF_3_) and 20 ppms (mono-fluoro-aromatics) [2], they may still be enough to characterize motional and structural properties of IMPs in different environments such as lipid vesicles, detergents or organic solvents.1D spectroscopy, used to avoid unfavorable relaxation associated with multidimensional NMR methods, is usually sufficient for the separation of the peaks in ^19^F spectrum (Figure 1A) and works even for potentially dynamic states that are characterized by broad lines. Thus, different states can be resolved, and their corresponding population quantified.Lastly, the lack of protein deuteration significantly improves the ease and efficacy of sample preparations.

The above advantages do come at a price. Assigning ^19^F resonances to structural states is anything but trivial and often requires prior hints from other techniques. For the thoroughly studied GPCR family (reviewed in [8]), the assignments rely on population shifts in response to agonists vs. antagonists or reverse agonists binding. In an impressive study from the Prosser lab [9], five key functional states of adenosine A_2A_ receptor (A_2A_R) complexed with hetero-trimeric G protein (Gα_s_β_1_γ_2_), were characterized in phospholipid nanodisc using ^19^F-NMR. Signal transduction was modeled using rigidity-transmission allostery (RTA) algorithms [10]; and A_2A_R conformational ensembles, corresponding to the above functional states, were visualized through the dynamic energy landscapes reflecting, activation, G protein coupling, and nucleotide exchange, as described in Figure 1B. 

**Figure 1 membranes-12-00227-f001:**
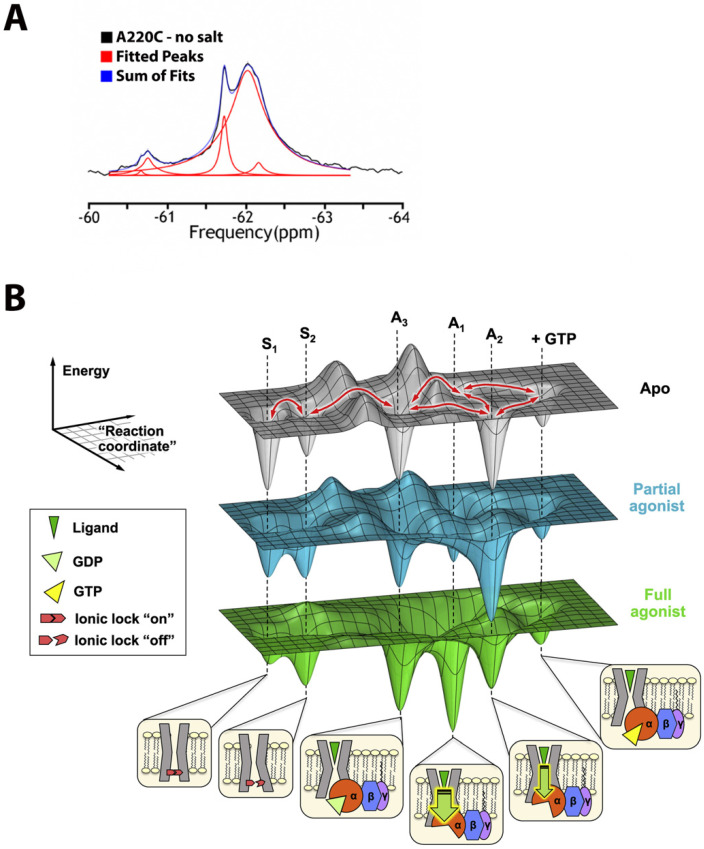
(**A**) Example of ^19^F-NMR spectrum, shown for NCX channel in nanodisc with ^19^F-probe attached to the side chain of residue 220. Signals are fitted by Lorentzian lines. (**B**) Dynamic energy landscape encompassing key functional states of adenosine A_2A_R [9]. The conformational ensemble is represented by five key functional states—two inactive states (S1 and S2) differentiated by the switching of a conserved ionic lock and three active states (A1, A2, and A3) associated with G protein coupling. A3, an intermediate or precoupled state, plays a role in the recognition and binding of the G protein. A1 and A2, on the other hand, are responsible for GDP release and stabilization of the nucleotide-free complex. While A1 is more efficacious (thicker downward arrow) and stabilized to a larger extent by the full agonist, A2 is less efficacious (thinner downward arrow) and is preferentially stabilized by a partial agonist. Reproduced with permission from Elsevier.

Alternatively, differences in paramagnetic relaxation enhancement (PRE) can be used for assigning ^19^F resonances to structural states [11], but only when different conformations, feature distinct solvent exposures for the ^19^F probe. In another novel approach, ^19^F longitudinal relaxation rates (R1) and their distance-dependent enhancement by paramagnetic ions (Ni^2+^ chelated through di-histidine motif) were measured with the goal to assign resonances to different structural states. This estimated the rates of the conformational exchange for a model membrane transporter—Glt_Ph_, aspartate/sodium symporter from *Pyrococcus horikoshii* [12]. 

The inter-conversion between different conformational states, occurring on a μs to ms time scale can be evaluated by Carr–Purcell–MeiBoom–Gill (CPMG) relaxation dispersion experiments [13]. This technique has been widely used to study dynamics and folding pathways in soluble proteins, for instance, to define “invisible” sparsely populated states [14]. In general, this method is not easily applicable to IMP/membrane-mimetic complexes of large molecular weight due to unfavorable relaxation rates in ^1^H-based multidimensional NMR experiments. The ^19^F-1D versions of these experiments, however, can be employed successfully to define exchange rates between GPCR states. For example, Manglik et al. showed that unliganded and inverse-agonist-bound β_2_-adrenergic receptor (β_2_AR) predominantly existed in two inactive conformations that exchange within hundreds of μs. Agonists shifted the equilibrium towards a conformation capable of engaging cytoplasmic G proteins, although incompletely, resulting in an increased conformational heterogeneity with the coexistence of inactive, intermediate, and active states [15].

In another interesting approach, remarkable relaxation properties of the aromatic ^13^C-F spin pair allowed for ^19^F-^13^C TROSY-based experiments, with direct detection from ^13^C. The 2D spectra produced sensitive, high-resolution signals for a target, such as the single-ring α7 proteasome particle with an MW of 180 kDa [16], demonstrating the potential of solution NMR to study IMPs.

Incorporation of extrinsic ^19^F-labels at relevant locations within the target protein is possible either through chemical conjugation of fluorine-containing small molecules with reactive amino acids or cysteine mutants, as was utilized in the above examples, or through biosynthetic introduction of fluorinated amino acid analogs [17,18]. An interesting example of the latter can be found in the study of β-arrestin-1, employing the unnatural amino acid (3,5-difluorotyrosine, F2Y) to define its V2Rpp bound (V2-vasopressin receptor carboxy-terminal phosphopeptide) conformation and dynamics [19]. As a result, a longstanding puzzle regarding the receptor’s phospho-coding mechanism that dictates selective structural features directed by either desensitization of the receptor or initiation of arrestin’s own signaling pathway was resolved. An elegant model was proposed which suggested that the phosphate-binding site on arrestin’s surface was arranged in “a shape similar to the holes in a flute” with movements controlled by the phospho-receptor “fingers”. 

## 3. Cryo-Electron Microscopy

Cryo-EM (cryo-electron microscopy), the most recent addition to the structural biology toolkit, can provide near-atomic resolutions of macromolecular assemblies. The advent of direct electron detectors, corrections for beam-induced movement and specimen drift along with developments in other auxiliary methodologies, has made cryo-EM the method of choice for membrane proteins, especially those recalcitrant to crystallization [20]. In contrast to X-ray crystallography, which relies on structural uniformity within the crystal lattice, single-particle cryo-EM offers the opportunity to inspect an ensemble of conformational states for dynamic systems, such as ion channels or cell surface receptors, snap-frozen in a thin layer of vitrified ice. Therefore, the steady-state distribution of different conformers under various buffer conditions or ligand titrations can be observed, elucidating potential pathways involving conformational rearrangements.

Hite et al. showcase this approach, presenting numerous states occurring along the reaction cycle of Slo2.2, a Na^+^-dependent K^+^ channel [21]. Series of datasets collected from a Na^+^ gradient concentration of 20–160 mM revealed an ensemble of closed structures with a minor population of an open conformer which increased with increasing concentration (Figure 2A,B). In contrast, a data set collected at a high concentration of 300 mM showed predominantly open conformers. Comparison of the titration conformers with activity data implied that not all open channels conduct ions. Interestingly, no stable intermediate structure was observed, suggesting that the channel opens in a highly concerted, switch-like process. This example highlights the importance of capturing images of a large enough number of single particles. By doing so, it becomes possible to define multiple sub-states within the major conformational states. 

From a historic perspective, the first-ever structure of an ion channel determined by cryo-EM with high enough resolution to allow identification of sidechains was TRPV1, a polymodal signal detector channel which belongs to a transient receptor potential (TRP) family. Liao et al. characterized the unligated closed state [22] while, in a companion study, Cao et al. defined two open states (toxin or capsaicin bound) [23]. These studies revealed that TRPV1 opening is associated with major structural rearrangements in the outer pore, including the pore helix and selectivity filter, and pronounced dilation of a hydrophobic constriction at the lower gate, suggesting a dual gating mechanism. Several high-resolution cryo-EM structures of the TRP family channels, incorporated into different membrane mimetics later followed [24,25,26,27]. More recently, Zhang et al. produced an impressive compendium of 25 structures of TRPV1 by adding a pair of toxins and/or altering the cations and pH [28]. The structures provided a mechanistic insight into the extensive pharmacology of TRPV1 by visualizing alterations in the selectivity filter and gate correlating their allosteric couplings with disparate stimuli. Conformational dynamics in TRPV1 have been further studied by TIRF microscopy [29] as briefly discussed further below.

Though membrane protein structural biologists rejoice at sidestepping the near-Sisyphean task of generating well-ordered crystals, cryo-EM still requires astute biochemical preparation of target proteins. Exemplifying this case, Chen and colleagues used a multifaceted approach, including disulfide trapping, X-ray crystallography, and cryo-EM, to isolate an elusive “open-channel” state of the prototypical elevator-type transporter Glt_Ph_ (Figure 2C) [30]. Glt_Ph_ is an archaeal homolog of excitatory amino acid transporters (EAATs) which mediate uptake of glutamate and aspartate in neuronal and glial cells. The EAATs maintain a peculiar side hustle, in that they also function as chloride channels and this activity is not coupled to amino acid transport [31]. This ion channel activity has been observed since the mid-1990s but was not structurally rationalized even through scrupulous cryo-EM examination of a solution ensemble of homolog Glt_Tk_ [32].

In vitro elucidation of the molecular bases of mechanosensation presents unique technical challenges and hearing is perhaps, at the biophysical level, the least understood of the senses. No fewer than three studies over the past year were released which employ cryo-EM to resolve states of prestin, an essential component of the cochlear amplifier [33,34,35]. Prestin is a member of the solute carrier 26 (SLC26) anion transporter family and functions as the electromotive molecule of outer hair cells (OHCs) which amplify incident sound waves. The membrane-integral motor pulls the unenviable double-duty of sensitivity to both membrane potential and tension [36]. By varying the anions and including the inhibitor salicylate, a collection of prestin states were captured revealing the protein as a domain-swapped dimer with an elevator-like conformational cycle. The structures indicated that prestin can tune its in-membrane cross-sectional area and (supplemented by molecular dynamics [MD] simulations) also locally alter its surrounding bilayer thickness. The studies unshroud the elegant means by which an integral membrane protein transduces an electrochemical signal into mechanical work. 

It is worth noting that despite the advances in cryoEM, high-resolution structural information from the flexible or dynamic parts of the protein remain elusive, a feature of prime relevance for single-pass IMPs. A recent remarkable study of integrin α_5_β_1_, in its apo and fibronectin bound forms, by Schumacher et al. [37] illustrates this point. The C-terminal portion of the receptor, including the flexible lower legs, transmembrane helices, and the cytoplasmic tails, remained unresolvable in cryoEM maps.

Observing long-lived, equilibrium structural states with cryo-EM is increasingly facile and, as with X-ray crystallography (described below), much effort is being spent on developing time-resolved variants to scrutinize short-lived intermediates. Sub-second time resolution has been primarily achieved in a rapid mixing step immediately prior to grid vitrification which can afford resolution of up to ~10 ms [38]. Rapid mixing is accomplished either through (i) a microfluidic setup prior to spraying the sample or (ii) spraying separate solutions directly onto a grid plunging into liquid ethane [39,40,41]. As a proof of concept for on-grid mixing, Dandey et al. observed large conformational shifts induced by rapidly mixing Ca^2+^ with MthK, a calcium-gated potassium channel which inactivates over the course of seconds [41]. Though the data were insufficient to clearly resolve an unseen open or intermediate MthK conformation, a time-resolved EM methodology by rapid mixing appears very feasible for membrane proteins with dynamics in the ms–s time scale.

The utility of cryo-EM is attested to by the burgeoning number of PDB depositions. New methodologies, however, must be cultivated for extracting dynamic structural information in the μs–ms regime where many protein motions lie. One attractive alternative strategy to microfluidic or on-grid mixing is flash photolysis of caged compounds to homogeneously initiate protein dynamism [42,43]. This approach was demonstrated using acid-sensing ion channel ASIC1a as a test subject, whereby protons were released from the photocaged sulfate MPNS to drop the solution pH and shift the channel from a resting state to desensitized.

## 4. Serial X-ray Crystallography

X-ray crystallography has historically been the gold standard for structure solutions with achievable resolutions occasionally extending below 1 Å. With improved detector speed and high photon flux at third-generation synchrotrons, time-resolved crystallography (TRX) became feasible in the 1990s. Though even with advanced facilities, a key difficulty to surmount in these experiments is homogeneous initiation of the reaction process, hence the targets typically studied are light-sensitive and can be triggered by a laser pulse. Many targets are, however, not light-sensitive but can be interrogated through clever application of photocaged substrates or shifts in temperature or pH. Another drawback of TRX is crystal packing itself which constrains protein conformation and may preclude some biologically relevant conformational changes. Nevertheless, the TRX approach is powerful in addressing ultrafast (fs–ps) motions as well as slower processes which do not grossly distort the protein and disrupt the crystal lattice.

TRX comprises two basic approaches: Laue diffraction methods on single large crystals and serial crystallography on many small crystals. The former relies on polychromatic (pink) X-ray beams and a full dataset can be collected using only a few diffraction frames, unlike with monochromatic X-rays. A primary weakness of Laue diffraction is, unfortunately, its intrinsic sensitivity to crystal mosaicity and stringent requirement for high-quality crystals. Serial crystallography was pioneered by X-ray free-electron laser (XFEL) sources and subsequently has been adapted for synchrotron sources as well [44,45].

Integral membrane proteins tend to be more refractory toward crystallization and thus few have been investigated by Laue diffraction methods. Bacteriorhodopsin (bR) is a well-studied protein, with the advantageous trait of comprising a light-sensitive retinal prosthetic group, but even its crystals proved too fragile to withstand repeated pump-probing [46]. The lone success has been with the photosynthetic reaction center complex from *Blastochloris viridis* [47,48]. The first study revealed no discernable conformational change upon photoactivation (3.3 Å resolution model) while the second showed reorientation of a tyrosine close to the “special pair” (SP) bacteriochlorophylls at a higher resolution of 2.9 Å.

The more fruitful avenue for TRX thus far has been through serial crystallography. Serial femtosecond crystallography (SFX) was realized at XFEL sources over the past decade to accommodate for ultrahigh photon flux which obliterates exposed materials via Coulomb explosion. Useful data can be collected via what is colloquially known as “diffraction before destruction”, whereby single diffraction frames can be detected (presuming a diffracting crystal is in the beam path) [49]. Sample destruction obviously necessitates a comparatively large quantity of experimental mass but, in practice, this downside tends to result in other gains in efficiency. For instance, [sub]micron crystals are less tedious to produce but can diffract well especially from the high intensity of an XFEL. Additionally, with time-resolved studies, small crystal size affords less optical absorbance (i.e., for stimulating laser pulses) and rapid reactant diffusion, both of which improve reaction synchrony across a crystal.

As with (Laue) crystallography, soluble proteins tend to be better studied by TR-SFX but the technique has been exploited for a handful of (primarily light-sensitive) membrane proteins. Photosystem I, in complex with electron acceptor ferredoxin, provided the first test case by observing alterations in diffraction intensities using laser pulse-probing at 5 and 10 μs delays [50]. The collected data were not sufficient for high-resolution model building but verified the utility of pulse-probe methods ported from Laue crystallography to TR-SFX. Photosystem II (PSII) is thus far the most thoroughly examined membrane protein by TR-SFX [51,52,53,54,55,56,57,58,59,60,61]. Despite its name, PSII is the first protein in the photosynthetic chain and catalyzes light-driven water oxidation at the oxygen-evolving complex (OEC), a Mn_4_CaO_5_ cluster [62]. The reversible photocycle of PSII has been proposed to encompass five redox states, S_0_–S_4_, which evolve over a μs–ms timescale after photon absorption [63]. Most of the TR-SFX studies showed subtle or no conformational alteration, though, efforts were hampered by the achievable resolution. Later works have focused on the PSII OEC and have successfully produced snapshots of intermediate states with oxygen and water coordination, bringing closer a detailed mechanistic understanding of photosynthetic splitting of water.

The dynamics of the photosynthetic reaction center were recently followed up using ultrafast (i.e., picosecond timescale) TR-SFX [64]. Using pump-probe delays of 1 ps up to 8 μs, the authors captured photoactivated structural perturbations of the aforementioned SP chlorophylls which lead to low-amplitude oscillations within the protein to accommodate charge transfer and heat dissipation.

Rhodopsins have been highly studied for decades and, besides being convenient model proteins, have become important biotechnological tools for optogenetics and synthetic biology. The pigment retinal is the rhodopsins’ essential cofactor which isomerizes upon photon absorption and induces conformational changes to elicit ion transport [65]. bR is, unsurprisingly, the model protein of choice and its dynamics have been probed from sub-picosecond through millisecond regimes [66,67,68]. The molecular movies derived through TR-SFX illuminate each step in the bR photocycle, directly visualizing retinal isomerization, side-chain reorientations, and backbone shifts which drive proton pumping (Figure 3). In addition, these structural data inform quantum mechanics/molecular mechanics (QM/MM) simulations and rationalize how the protein scaffold has been evolutionarily honed to optimize the retinal quantum yield. Subsequently, the maturing TR-SFX method was directed on rhodopsins KR2, ClR, and ChR to decipher the physicochemical origins of their differing substrate selectivities [69,70,71].

Rounding out the membrane proteins subjected to TR-SFX is bovine cytochrome *c* oxidase (C*c*O) [72,73]. C*c*O catalyzes the reduction of dioxygen to two water molecules and is the terminal enzyme of the electron transfer chain in the inner mitochondrial membrane. To induce previously unseen intermediates, two studies used disparate stimuli: CO photolysis or rapid mixing with oxygen-saturated buffer. Both studies observed a similar oxidized heme *a*_3_ intermediate and, complemented with TR spectroscopy, gleaned new details on the enzyme’s trafficking of electrons to effect unidirectional proton flux. These subjects highlight the strength of TR-SFX in resolving protein structural dynamism in the fs to ms timescale. User access to XFEL sources is, unfortunately, very limited but facilities are continuously innovating to increase their data acquisition and analysis capabilities [74].

## 5. Förster Resonance Energy Transfer

Förster Resonance Energy Transfer (FRET) is a coulombic mechanism of energy transfer, useful for measuring distances on a molecular scale. It relies on electrical dipole moments and the transfer of the energy associated with an oscillating dipole which occurs on molecular excitation [75]. Photoexcitation during FRET induces an oscillating dipole in the excited molecule providing the transition dipole moment for the energy donor. A nearby molecule perceives the oscillating electric field associated with the donor’s oscillating dipole, inducing a dipole moment in the acceptor. Energy transfer occurs when the donor settles back into its ground state and the acceptor now oscillates with a transition dipole moment [76]. Effectively, the oscillating dipole in a donor induces an oscillating dipole in the acceptor. The change in energy associated with the relaxation of the donor is equal to the change in energy associated with the excitation of the acceptor. This common energy state depends on spectral overlap: for any donor and acceptor FRET pair, there needs to be an overlap between the emission spectrum of the donor and the absorption spectrum of the acceptor (Figure 4A). One can detect smaller differences in distances coming from differential FRET responses between donors and acceptors. The rate of energy transfer is defined by the equation:k_ET_ = α(κ^2^k^0^_D_/r^6^_DA_)*J*ε_A_(1)
for high FRET, the donor (D) and acceptor (A) should have a large spectral overlap integral *J*; high radiative rate k^0^_D_ and absorptivity ε_A_; and both D and A should be in near proximity to each other, with r_DA_ distance usually within 1 to 10 nanometers. For biological applications the term R_0_ is defined as the distance at which the FRET efficiency is 50% (Figure 4A) and the overall equation follows [77]:E = 1/(1 + (r/R_0_)^6^)(2)

For time dependent analysis
k_ET_ = (R_0_/r)^6^(1/τ_D_)(3)
where τ_D_ is the donor’s fluorescence lifetime in the absence of the acceptor.

FRET approaches have gained widespread popularity in recent years with the availability of fusion proteins such as green fluorescent protein (GFP) and/or its mutated derivatives. FRET is often used to measure variations in intra- and inter-molecular distances occurring, for e.g., during opening of an ion channel or changes in the activation state of a cell surface receptor.

Single-Molecule FRET

With the emergence of single-molecule detection [78,79], FRET has become a powerful tool to study intrinsically disordered proteins (IDPs) [80,81], protein folding processes [82,83], and other structure-function dynamics [84,85]. Single-molecule FRET (smFRET) [86] can provide molecular insights with a high spatial resolution of 2–10 nm in an accessible time scale of ns-minutes [87]; structural changes, which otherwise would be concealed by ensemble averaging of structurally heterogeneous sub-states can now be identified. smFRET is carried out on molecules that either diffuse freely or are immobilized to a surface [88]. In the context of membrane proteins, the labeling efficiency and stability within an appropriate membrane mimetic are of paramount importance. Typically, cysteine is the de facto residue of choice for attaching a fluorophore, while the endogenous cysteines in the protein are mutated out for an unambiguous study [89]. A common workaround is through orthogonal labeling with unnatural amino acids at the site of investigation [90]. Purified membrane proteins can be obtained through heterologous overexpression and purification in detergents [91] or incorporated into liposomes [92]. Membrane proteins reconstituted into nanodisc provide a detergent-free system [3,93] for applications requiring a two-dimensional membrane (Figure 4B). TIRF (discussed below) and confocal microscopy are commonly employed in the detection of smFRET [94]. 

The most prominent examples of FRET come from classes of membrane proteins called transporters and ion channels where a multitude of data has emerged from site-specific labeling on either side of the membrane and monitoring their rearrangement in real-time. We briefly discuss a few examples to highlight the application of FRET. 

MsbA is an adenosine triphosphate (ATP)-binding cassette (ABC) transporter from gram-negative bacteria, and is critical for flipping Lipid A using energy from ATP hydrolysis. ABC transporters consist of two transmembrane domains (TMDs) responsible for substrate selectivity and two cytoplasmic nucleotide-binding domains (NBDs) that catalyze ATP hydrolysis. smFRET analysis of MsbA using probes on the cytoplasmic and periplasmic sides identified inward-open, intermediate, and outward-open states for the cytoplasmic face, while the periplasmic face was devoid of an intermediate state (Figure 4C(i)). MsbA was then analyzed to determine the effect of different membrane mimetics. In nanodisc, the NBDs were close to each other in the apo form, while in the presence of ATP the domains moved even closer registering higher FRET. In sharp contrast, in both liposomes and detergents, the apo state showed a distinct inward open state, with the NBDs separated, followed by a closed state in the presence of ATP. Conformational dynamics in the TMDs were determined by periplasmic FRET. In nanodiscs and liposomes the apo form assumed a periplasmic closed conformation which became less compact on binding to substrate as observed from slightly lower FRET, hinting at a lower conformational transition. In detergent, the apo form showed the less compact periplasmic FRET, but transitioned to a periplasmic open conformation upon addition of substrates [95]. Altogether, they showed that the dynamics across the membrane was different for either side, dependent on the mimetic system, and qualitatively coupled. 

DtpA, found in *E. coli.*, belongs to the family of proton-dependent oligopeptide transporters (POTs). They have roughly twelve transmembrane helices organized in two six-helix bundles [96]. FRET studies of DtpA in detergent revealed that the protein had a flexible conformational ensemble of inward-open states suggesting variations in the extent of cytoplasmic opening. For the same FRET pair, placed at the cytoplasmic side, a small population of high FRET was observed indicating either an outward-open conformation or an occluded state. To differentiate between the two, a conformation-specific nanobody against a closed periplasmic structure (inward open conformation) was employed that would trap the protein in an inward-open state leading to a low FRET signal. Remarkably, the nanobody did not induce the intended effect proving the presence of an occluded state with simultaneously closed cytoplasmic and periplasmic sides (Figure 4C(ii)). An occluded state intermediates between the transfer from an outward to inward-open conformation. Furthermore, Lasitza-Male et al. investigated the role of lipids in capturing the elusive outward-open state. Detergent-free preparation of DtpA in Saposin nanoparticles (SapNP) [97] in the presence of different lipids were explored. With POPE (1-palmitoyl-2-oleoyl-sn-glycero-3-phosphoethanol-amine) lipid there was an increase in the sampling of outward-open state and a perceptible effect in dampening of FRET in the presence of nanobody. Nevertheless, other conformers were also equally observed. Further experiments with POPS (1-palmitoyl-2-oleoyl-sn-glycero-3-phospho-l-serine), POPA (1-palmitoyl-2-oleoyl-sn-glycero-3-phosphate), and BL (brain lipid) extract accentuated the importance of lipid head groups, where POPA and POPS led to the detection of a more extended inward-open conformer. Interestingly, BL extracts exclusively showed inward-open conformer [98]. The smFRET study on DtpA highlights the importance of lipids in understanding the cooperativity between the cytoplasmic and periplasmic sides of a membrane protein.

Neurotransmitter:sodium symporters (NSSs), in the SLC6 family, are secondary active transport proteins that, coupled to cellular sodium gradients, reuptake neurotransmitter molecules following their release into the postsynaptic space. Several psychostimulants and antidepressants act by influencing this family of symporters [99]. The smFRET studies on LeuT, a bacterial NSS homolog, shed important light on the dynamics of molecular events in the transport mechanism of substrates across the membrane. Terry et al. demonstrated weaker coupling between the intra- and extracellular sites and identified transitional states by observing intermediate conformations for both the extra- and intracellular regions. The authors also found a preferential abundance of the partially inward open intermediate in the presence of substrate and Na^+^ ions that was reversed in the presence of high Na^+^ concentration and absence of substrate [100]. Several structures of LeuT bound to substrates with different transport rates highlighted the importance of their interaction with residue F259 in TM helix 6b (PDB 3GJD) (Figure 4C(iii)). A follow-up study by LeVine et al. using smFRET in combination with MD simulation demonstrated that residues F259 and I359 elicits substrate-specific allosteric modulation of gating and Na^+^ release. F259 acts as a volumetric sensor for the size of a substrate. Low-volume substrates induce rapid LeuT sampling of an inward open intermediate while allowing free rotation of F259, while bulky substrates prevent rotation and the transporter samples an outward open conformation [101]. Residue homologous to F259 in dopamine, norepinephrine, and serotonin transporters could be expected to play similar roles.

Fluorescent unnatural amino acids (fUAAs) are gaining traction as an alternative to site-specific thiol labeling. fUAA is genetically encoded by an *amber* nonsense codon (TAG) and, with the help of orthogonal fUAA-tRNA^CUA^/tRNA-synthetase pair, is introduced into the protein. A major advantage lies in the avoidance of endogenous cysteine that may have functional relevance; or circumventing a plethora of cysteines that hinders site-specific labeling. The latter was a problem in studying voltage-gated potassium channels (K_V_) using voltage-clamp fluorometry (VCF). K_V_ channel is a tetramer with TM helices S1–S4, forming the voltage-sensing domain, and S5-S6 forming the pore domain. Using fUAA, 3-(6-acetylnaphthalen-2-ylamino)-2-aminopropionic acid (Anap), at certain positions the authors showed that independent conformational changes occur at the N- and C-termini of the S4 helix followed by a cooperative two-step transition at the S6 helix leading to a global rearrangement in the protein [102]. Another study using the unnatural amino acid p-acetyl-L-phenylalanine (AcF) was done with the GluN1 agonist-binding domain (GBD) of N-methyl-D-aspartate (NMDA) receptor. Removal of endogenous cysteine was detrimental to the protein’s expression and hence smFRET studies were carried out using unnatural amino acid. The study showed that the efficacy of the agonist was based on conformational selection rather than the formation of intermediate states, and there was a preferential spread for conformational cleft closures in the presence of agonist, partial-agonist and antagonist [103]. An innovative application in smFRET has been demonstrated by caged fluorescent dyes, a novel class of fluorophores that can be converted from an inactive nonfluorescent state to emit fluorescence on activation by reductive reagents or photoactivation by UV light [104]. A prototypical application of using caged chromophores has been demonstrated for the BetP transporter [105]. Self-healing fluorophores is another technological advancement where a more soluble and potentially non-toxic fluorophore is chemically engineered with protective agents 1,3,5,7-cyclooctatetraene (COT), 4-nitrobenzylalcohol (NBA) and Trolox. Conjugation to Cy5 showed increased effective local concentration, little blinking and reduced photobleaching rates when compared to their unconjugated parent [106]. Self-healing fluorophores enable, high brightness, signal-to-noise ratio, and total photon counts that facilitate imaging at faster temporal resolution. Such fluorophores have been used in the study of dopamine receptor D2 [107].

Total Internal Reflection Fluorescence

Total Internal Reflection Fluorescence (TIRF) microscopy utilizes the property of an induced evanescent wave at the interface between two media having different refractive indices which results in its confinement to the higher index medium. As a result, specific fluorescent excitation can be achieved in a very thin (usually <100 nm) optical section. This eliminates background fluorescence from outside the focal plane, thus improving the signal-to-noise ratio and spatial resolution of the measurement. This approach is perfectly suited for fluorescence-based studies of membrane proteins.

Steinberg and coworkers [29] took advantage of TIRF microscopy, to study TRPV1 and image conformational dynamics in live cells using the fluorescent amino acid coumarin-tyrosine. Y671, a residue proximal to the selectivity filter was chosen as the site for incorporating the fluorescent probe. They showed that photon counts and optical fluctuations from coumarin encoded within channel tetramers correlate well with activation by capsaicin, measured in whole-cell macroscopic electrophysiological recordings, thus providing an optical marker of conformational dynamics at the selectivity filter. This is a good example of establishing the correlation of single-molecule dynamic measurements with functional data originating from averaged ensembles. The experimental data was further supported by MD simulation suggesting a divergent solvent exposure of Y671 in the closed and open states. Overall, the authors proposed a connection between the dynamics of the selectivity filter and the mechanism of gate permeation.

## 6. Electron Paramagnetic Resonance 

Electron paramagnetic resonance (EPR) is a spectroscopic technique that can be applied to molecular systems with an unpaired electron, as opposed to nuclei in NMR. Sometimes also referred to as electron spin resonance (ESR), it utilizes microwave (MW) radiation that probes paramagnetic centers and radical cofactors in the presence of an external magnetic field (B_0_) [108]. Depending on the orientation of angular momentum, electrons have degenerate spin states in the absence of an external magnetic field. 

In the presence of a magnetic field the spin states are separated by energy known as the Zeeman effect (Figure 5A):ΔE = hυ(4)

The EPR spectrometer provides a linear sweep of a magnetic field while exposing the sample to a fixed frequency of microwave radiation known as the continuous wave EPR (CW-EPR) [109]. Of the several available frequencies, X band (~9.6 GHz) is most popular though L and Q bands are also used for CW-EPR. An EPR spectrum is displayed as the first derivative of the absorption spectrum observed when the resonance conditions are met on sweeping with B_0_. The frequency for which the resonance condition is fulfilled is called the Larmor frequency. The field position of the resonance can also be determined from the precisely known g-factor for electron using the equation:g = hυ/μ_B_B_0_(5)
(h = Plank’s constant; υ = frequency; μ_B_ = Bohr magneton) 

Various nitroxide (-NO) spin labels are used in EPR studies viz., (a) methanethiosulfonate spin label (MTSL), (b) maleimide spin label (MSL) N-(1-oxyl-2,2,6,6-tetramethyl-4-piperidinyl) maleimide, (c) iodoacetamide spin label (ISL), (d) bis(1-oxyl-2,2,5,5-tetramethyl-3-imidazolin-4-yl) disulfide (IDSL), (e) bifunctional spin label (BSL), (f) 2,2,6,6-tetramethyl-N-oxyl-4-amino-4-carboxylic acid (TOAC), and (g) 4-(3,3,5,5-tetramethyl-2,6-dioxo-4-oxylpiperazin-1-yl)-l-phenylglycine (TOPP) [110]. The electron magnetic moment from the unpaired electron, shared between the O and N atoms, interacts with the nuclear magnetic moment of nitrogen (known as the hyperfine interaction) and leads to spectral splitting (Figure 5A). This splitting follows the rule of multiplicity:M_I_ = 2I + 1(6)
(I = nuclear spin number, in this case 1).

Multiple strategies are available to incorporate a spin-label including heterologous expression of a protein or chemical synthesis of a peptide followed by site-specific reaction of the spin-label through a disulfide linkage; or use of a non-covalent label that can bind to a hydrophobic pocket within a protein. If the cysteine residue is functionally important, alternative strategies using unnatural amino acids labeled with a nitroxide spin label can be employed [111]. The EPR spectra obtained from a spin-labeled protein when compared to the spin-label itself differs in the line width of the hyperfine peaks due to peak broadening and amplitude dampening arising from a decrease in the orders of freedom experienced by the spin probe (Figure 5A). The line shapes of EPR can either be affected by the dynamics of the system or by spin-spin interaction where two spin labels are in close proximity to each other. The latter is utilized, in frozen samples, to determine short-range conformational changes by comparing EPR spectra of a double-labeled sample with a single-labeled protein. Dipolar broadening is observed if the spin labels are 8–25 Å [112], beyond which no spin couplings are detected. This shortcoming was addressed by the development of pulse-EPR. 

Pulse-EPR spectrum is recorded by exciting a large frequency range simultaneously with a single high-power MW pulse of frequency υ at constant B_0_ [113]. Here a small segment of the CW pulse is taken and amplified to obtain a pulse length in the time domain which is converted to the frequency domain by Fourier transformation giving the excitation bandwidth. Pulse-EPR can measure spin echo amplitude as a function of various quantities such as field, pulse delay, RF frequency and MW frequency. Double Electron-Electron Resonance (DEER) or Pulsed Electron Double Resonance (PELDOR) is an example of measurements done as a function of MW frequency obtaining long ranges distance (~2–10 nm) between two spin states connected through space [114]. Dipolar coupling distributions measured by DEER employ a four-pulse DEER sequence which consists of a refocused primary echo at the observer frequency ωA, followed by a time varied inversion pulse at the pump frequency ωB producing a refocused two pulse echo [115] (Figure 5B). Of the several conditions that improve the signal to noise ratio of DEER experiments, is the use of deuteration, which analogous to NMR, reduces non-specific dipole-dipole couplings and allows for the increase of t_max_ (maximum dipolar evolution time) to measure longer distances, up to 140 Å [116]. The refocused echo shows modulations in the time domain that provides information on dipolar-dipolar coupling between the two interacting spin systems. The time-domain data F(t) can be converted to frequency domain υ_dd_ using Fast Fourier transformation and used to determine distance from the Pake pattern since the dipolar-dipolar frequency is inversely proportional to distance between the spins (r^3^); alternatively, a model-independent analysis can be carried out to directly convert the time domain data into distance information P(r). Transformation of F(t) to P(r) is a moderately ill-posed problem that is addressed by distance domain smoothing through Tikhonov regularization [117]. Finding the optimal regularization parameter (α) in the L curve provides the optimal probability of distance distribution P(r). Traditionally, DEER measurements are carried out the at X band but there is now a growing trend moving towards the use of Q band to increase sensitivity [118]. It is important to note that dynamic information can be gleaned from room temperature experiments done in solution while distance measurements are done on frozen samples. 

EPR analysis can generally provide information, on the mobility of the sidechain modified with the spin label; the accessibility of a hydrophobic or hydrophilic environment experienced by the sidechain; and the relative distance between two spin labels in a protein using DEER/PELDOR. The technique is not limited by the size of the protein and can detect distance measurements of <170 Å, as has been demonstrated for a large protein complex GroEL using Q band and deuteration [119]. It is important to note that such large distance measurements are more of an exception that routine. Often during a protein’s activation cycle, the rigid structure undergoes remarkable conformational changes that can be deduced using techniques such as EPR. The following examples highlight the importance of EPR in the context of understanding a protein’s function. 

β_2_AR is a GPCR involved in signals transduction across the membrane. In one study by Manglik et al. [15] 3-(2-iodoacetamido)-2,2,5,5-tetramethylpyrroline-1-oxyl (IA-PROXYL) was placed at the cytoplasmic end of TM 4 and 6 in β_2_AR purified in detergent molecules (Figure 5C(i)). The conformational rearrangements were monitored in the presence of low- and high-affinity inverse agonists, agonists and agonists with nanobody (Nb80). DEER experiments in the presence of inverse agonists showed that TM6 which undergoes large conformational change upon receptor activation, demonstrated heterogeneity in the inactive state. A feature previously unobserved in crystal structures (see section on antibody fragments). Typically, in the inactive state, an “ionic lock” is formed between the DRY motif in TM3 and E285 in TM6. The heterogeneity from the DEER experiments hinted towards two different, equally present, inactive states S1 and S2 with the ionic lock broken in the latter. The apo receptor also showed both states, albeit with a lower rate of interconversion. A high-affinity agonist with Nb80 (a G protein surrogate) produced a single conformational state S4, stemming from the large rearrangement of TM6, breaking the ionic lock. Interestingly agonists with no Nb80 produced a close but distinct conformation S3, a possible intermediate between the inactive (S1, S2) and active state S4. This subtle conformational difference evaded detection by DEER but was picked up through NMR. Remarkably, both agonists also showed a substantial population of receptor in S1 and S2 inactive states. The broken ionic lock S2 might be a critical state in overcoming the energy barrier towards the formation of an active intermediate S3. On β_2_AR activation, the heterotrimeric G protein engages with the S3 state and promotes the formation of a stable complex (S4). The results highlight the dynamic nature of β_2_AR activation that interconverts between various active and inactive states. Intriguing DEER measurements of β_2_AR have also been conducted under the influence of applied hydrostatic pressure to reveal structural features of sparsely populated conformers [120]. 

EPR studies are typically done on membrane proteins purified in detergents and reconstituted in lipid bilayer, typically liposomes. Spin-labeled cysteine mutants of sodium/aspartate symporter Glt_Ph_ were used to investigate distance measurements in the apo, and substrate-bound form [121]. Glt_Ph_, a homo-trimer, was labeled at its cytoplasmic and extracellular domain along with the HP2 hairpin loop which serves as an extracellular gate. Broad distance distributions reminiscent of independent protomers sampling inward- and outward-facing states were observed in detergent and liposomes. The observation of shorter average distances in lipid bilayer than detergent suggested a more compact structure hinting towards the effect of lipids on the energetics of membrane proteins. These results were further independently corroborated by Hanelt et al. who also conducted similar EPR studies [122]. 

Pentameric ligand-gated ion channels (pLGICs) mediate synaptic transmission by opening or closing the channel in response to a neurotransmitter. The inactive channel is either in a desensitized state while still bound to a neurotransmitter, or in a closed resting state. Basak et al. [123], showed the effect of docosahexaenoic acid (DHA), a lipid molecule found in the neuronal membrane, on its ability to desensitize GLIC, a proton-gated bacterial ion channel. GLIC has four TM helices (M1–M4) followed by an extra-cellular domain. The crystal structure of GLIC-pH4-DHA (PDB 5J0Z) showed that DHA is bound in the vicinity of the outermost helix M4, an established lipid sensor. EPR measurements were conducted to probe the conformational changes in M4, since it was unexpectedly rigid upon comparison with the structure of GLIC-ph4 (PDB 4HFI). CW-EPR of residues along the M4 helix showed increased mobility of residues on moving away from the intracellular end. The flexibility was preferentially more for residues facing the pore region indicating a disruption of M4 association with M3 and M1. GLIC was reconstituted in nanodisc, since a longer distance measurement using DEER is cumbersome in liposomes. Q band data collection in the closed and desensitized state showed the presence of relatively longer distance measurements indicating the outward movement of M4 helix (Figure 5C(ii)). A comparison of CW spectra at the M4 position, in the presence and absence of DHA, in conditions of neutral and acidic pH proved that DHA indeed stabilized an agonist-induced desensitized state. 

Another remarkable study by Mishra et al., deducing the conformational cycle for the heterodimer BmrCD was achieved exclusively using DEER spectroscopy [124]. Spin labels were placed at strategic positions on the extra- and intracellular interface and the intracellular NBD for each protomer BmrC and D. Experiments were conducted in the presence of substrate, AMP-PNP (non-hydrolysable ATP analog), and ADP-Vi (mimics transition state of ATP hydrolysis or HES). The apo state showed large distance distribution representing disengaged NBDs, which reduced in the presence of AMP-PNP and ADP and narrowed further with ADP-Vi forming a closed NBD sandwich. Both AMP-PNP and ADP stabilized an intermediate conformation that nucleates the formation of NBD dimers. Spin label pairs at the extra- and intracellular interface showed the transition of TMD from an inward to outward facing conformer upon ATP hydrolysis. Positions of the catalytically conserved motifs in the nucleotide-binding site (NBS) are essential for ATP hydrolysis. An overlapping distribution of nucleotide bound and hydrolyzed states were observed for the consensus NBS (cNBS), indicating the formation of HES population upon ATP binding. Distinct distributions were observed for the degenerate NBS (dNBS) suggesting the need of ATP hydrolysis to observe HES. All the above conformational states were retained in a nanodisc further accrediting their bona fide existence. Under turnover conditions in the presence of ATP and substrates, the conformational population shifted from HES to nucleotide bound state for dNBS while for sNBS an additional apo state was observed. Thus, the two NBSs remained in mutually different catalytic intermediates while cycling through conformational states revealing structural asymmetry (Figure 5C(iii)). Altogether, the study parsed out conformational flexibilities undertaken by the various domains in the heterodimer. 

In addition to purifying membrane proteins in detergents, followed by reconstitution into membrane mimetics such as proteoliposomes or nanodiscs, efforts are also being made to understand conformational changes of membrane proteins in cellular environments [125]. BtuB is an outer membrane cobalamin transporter in Gram-negative bacteria. It is a 22-stranded β-barrel protein with an N-terminal plug or hatch domain in the center. The protein was labeled, by adding MTSL to the cell suspension, at two engineered cysteine residues facing the extracellular (EC) surface. Molecules less than 600 Da can transverse the OM and enter the periplasm. MTSL remains reduced in the periplasm and cannot provide a signal that would interfere with extracellular EPR measurements. Room temperature CW-EPR measurements from labeling the hatch domain at two cysteine residues showed that the N-terminus was static in the cellular environment discrediting the hypothesis of a domain rearrangement. Additional labeling of two loops (one flexible and the other rigid) on the β-barrel EC region showed an expected mobile spectrum from RT CW-EPR. DEER measurements in the apo state showed a broad conformational distribution which converted to a distinct conformation upon addition of Ca^2+^, a feature demonstrated by the visibility of the flexible loop in crystal structures of the apo/apo- Ca^2+^ state. Upon further addition of the substrate cyanocobalamin, the mean distance increased indicating a general effect of the substrate. The use of the bi-specific label could ameliorate the observation of broad distance distributions. Similar measurements done with isolated membranes demonstrated a small difference in distance measurements but retained an overall similar effect of ligand on loop movements as observed in cellular environments. Spin labels such as nitroxides, shielded nitroxides and Gd^3+^, are popular with in-cell EPR applications [126]. 

## 7. Hydrogen-Deuterium Exchange 

A combination of mass spectrometry (MS) with hydrogen-deuterium exchange (HDX) or hydroxyl radical foot-printing (HRF) has been in use for assessing structural and dynamic properties of purified proteins since the 1990s. The two major methods of foot-printing (reviewed in [127])—reversible labeling of backbone amides through HDX, and irreversible covalent labeling of side chains by HRF—were initially applied to soluble proteins and have been extended to membrane proteins for defining protein interfaces, mapping ligand binding sites, monitoring dynamics of conformational changes, etc. Limited quantities of sample requirement with the ability to run experiments in solution, at room temperature, in several membrane mimetics, are among the major advantages with the caveat of possible incomplete sequence coverage.

In HDX-MS [128,129], stability of hydrogen bonds in the backbone, amides are assessed through the measurement of isotopic exchange between hydrogen and deuterium. Amide hydrogens, which participate in stable hydrogen bonds of secondary/tertiary structures, or are solvent inaccessible within the hydrophobic core, are protected and unexchangeable. Labeling occurs by protein dilution or rapid buffer exchange into pure D_2_O to produce >95% deuterium concentrations. Aliquots of the reaction are quenched with acid at different time intervals, digested by proteases, and analyzed by MS. Peptide mass values and intensities are extracted and a time course of the increase in mass is obtained. 

In HRF-MS [130], among the methods tested to generate suitable hydroxyl radicals, a few that have found practical applications include: radiolysis of water with electrons, X-ray or gamma radiation; photolysis of peroxide; transition metal-dependent chemical reactions with peroxide; and high voltage electrical discharge in water. Proteins are exposed to hydroxyl radicals for selected time intervals and quenched by addition of quenchers (for e.g., 10 mM methionine amide), followed by MS. Since oxidative modifications are stable and irreversible, the downstream analysis for HRF-MS is more flexible than HDX. 

NhaA protein from *E. coli.*, a model representative of the Na^+^/H^+^ antiporters family, has been studied by HDX-MS in detergent to analyze conformational changes upon Li^+^ binding at physiological pH [131]. The antiporter has two domains, the interface domain and the Li^+^ binding core domain. The observed differences in deuterium uptake indicated that Li^+^ induced two conformational rearrangements along the cytoplasmic and periplasmic funnels. Regions around the residues directly involved in Li^+^ binding remained unperturbed while the most pronounced effect was seen in loops that connected the two domains (Figure 6A). Effectively, a mechanism was proposed where an immobile binding site is exposed to either side of the membrane by a translational motion of the core domain supporting the alternating-access mechanism. 

The membrane-bound molecular motor F_0_F_1_-ATP synthase was studied using HDX-MS in *E. coli.* membrane vesicles [132]. This protein is a multimeric complex of several components where the γe-rotor shaft connects α3β3 with the ac_10_ proton translocator. The study was carried out under three experimental conditions of ADP-inhibited and catalytically active states in the presence and absence of proton motive force (PMF). Analysis of peptides from deuterium exchange identified a C-terminal helix in the γ subunit that revealed structural perturbations upon rotational catalysis in the presence of PMF (Figure 6B). This effect was absent in the stationary F_0_F_1_ or upon rotation in the absence of PMF. The rate of deuterium exchange showed that this region was ordered with transient opening/closing of the H-bonds. The energy calculated from the destabilization of H-bonds reflected accumulated torsional stress in γ that builds up while F_0_F_1_ is catalytically active. Effectively γ acts as an energy reservoir that prevents the return to a torsionally relaxed conformation between power strokes. 

LeuT, a neurotransmitter:Na^+^ symporter (NSS) has been extensively studied to understand the conformational mechanisms of biogenic amines transport. Extensive structural studies of LeuT, predominantly performed on detergent-solubilized protein, suggested a large-scale structural rearrangement between outward- (OF) and inward-facing (IF) conformers [133,134,135]. The HDX-MS investigation, conducted in nanodisc, supported the idea of a transition from OF to IF to involve the occlusion of the extracellular vestibule and opening of the cytoplasmic pathway, the latter being partly facilitated by the untethering of the N-terminal peptide and TM1a from the scaffold [136]. In another study by Reading et al. SMA-nanodiscs with varying lipid compositions and HDX-MS were used to investigate the importance of lipid environment in modulating conformational dynamics in membrane proteins. *E. coli.* rhomboid protease GlpG is a ubiquitous family of intramembrane serine proteases that cleave peptide bonds within the lipid bilayer. Changes in accessibility and dynamics of GlpG, captured within three different native lipid compositions, were demonstrated, along with the identification of protein regions, sensitive to variations in lipid environment [137]. 

Interestingly, HRF-MS has been applied, although with very few examples, to study membrane proteins in living cells. Zhu and co-workers [138] were among the first to investigate the outer membrane porin OmpF using modified Fenton chemistry for in situ radical generation:Fe(II) + H_2_O_2_ → Fe(III) + •OH + OH^−^(7)

Oxidatively modified peptides from the open, but not closed, channel, were identified in the loop regions and in β-strands, demonstrated gating by OmpF in a cellular system. Another method for generating hydroxyl radicals is photolysis of hydrogen peroxide by laser:*hv*
H_2_O_2_ → 2•OH(8)

This HRF approach is also known as “fast photochemical oxidation of protein” (FPOP). The nanosecond laser photolysis, followed by protein enrichment through immunoprecipitation and LC-MS analysis, was utilized to study the interactions between epidermal growth factor (EGF) and its receptor (EGFR) in live mammalian cells [139]. Overall, the data suggested that upon interaction with EGF the receptor assumed an extended conformation. 

## 8. Antibody Fragments

Antibody engineering has become a fundamental field in the effort towards fighting long chronic diseases such as cancer [140]. The mAb (monoclonal antibody) technology makes the bulk of this effort with more recent forays into engineering smaller antibody fragments [141]. The motivation for developing antibody fragments stems from an improved rate of tumor uptake and intratumoral distribution [142]. Besides clinical applications, mAb and antibody fragments have proved revolutionary in the field of scientific research with applications in protein detection [143], immunohistochemistry [144], ELISA [145] and membrane protein structure determination [146]. Antibody fragments generally minimize conformational heterogeneity and intrinsic flexibility along with improving crystal contacts leading to better X-ray diffraction and high-resolution structure determination of membrane proteins [147,148]. They additionally act as fiducial markers in cryo-EM applications [149].

The basic structure of an antibody consists of four polypeptide units with two identical heavy and light chains held together by disulfide bonds. With an overall Y shape, the light chains are shorter than the heavy chains. Each polypeptide chain has a constant (C) and variable (V) region. The variable regions interact with specific antigen and is responsible for the antigenic diversity. The heavy/larger chain contains three constant regions with C_H_2 and C_H_3 located below the flexible hinge, while C_H_1 extends into the variable V_H_ region above the hinge. Conversely, the light/smaller chain only contains a single constant C_L_ region that extends into the V_L_ variable region, all present above the hinge (Figure 7A). A conventional antibody can be subdivided into different sections: fragment variable (Fv) region, which contains the two variable domains V_H_ and V_L_; fragment antigen-binding (Fab) region, which includes the constant and variable domains of both the heavy and light chains above the hinge (C_H_1-V_H_ and C_L_-V_L_); and fragment crystallizable (Fc) region, which exclusively consists of constant regions from the heavy chains (C_H_2-C_H_3). Antigen binding fragments can be produced from protease and chemical digestion of conventional mAb. Digestion of mAb with the enzyme papain can produce individual Fab fragments with monovalent antigen-binding sites, while divalent fragments (Fab’)2 with an intact hinge region can be obtained with pepsin digestion. The smallest Fv fragment can also be produced by enzymatic cleavage. The single-chain variable fragment (scFv) is created by linking the Fv regions with an engineered peptide linker (Figure 7A) [149].

Camelids and sharks produce antibodies that are exclusively composed of heavy chain homodimers called heavy chain antibody (hcAb). The Fv portion of hcAb is called a nanobody, denoted as VHH in camelids and VNAR in sharks, and is the smallest naturally occurring antigen binding domain (Figure 7B). The variable region in camelids have three distinguishing features with respect to humans: (1) they have a single domain; (2) one of the complimentary determining regions (CDR3) in VHH is much longer than that of human V_H_ (the CDR3 loop contributes the most with respect to the diversity and specificity of the paratope); and (3) human V_H_ domain contains a hydrophobic region that interfaces with the V_L_ domain while the VHH domain is completely hydrophilic [150]. Because of their lack of glycosylation, increased stability, and smaller size nanobodies can easily be recombinantly produced and purified from prokaryotic expression systems. Hence, they have found wide acceptance in several biochemical, biophysical [151] and structural biology applications of membrane proteins [147,148]. Typically, a llama is sequentially immunized subcutaneously with purified membrane protein, followed by consecutive rounds of panning by phage display using the purified protein as bait. The positive clones are sequenced isolating unique VHH binders. Subsequently, the best binder is heterologously overexpressed and complexed with the purified [152] membrane protein. Recent efforts to circumvent the immunization of animals have led to the development of synthetic libraries that are displayed on the surface of yeast [153] or ribosome [154]. Additionally, several non-antibody molecular scaffolds are also available as synthetic binders to effectively reduce conformational flexibility in membrane proteins [155].

Antibody fragments have been instrumental in capturing these various conformations associated with a receptor’s transition from an inactive to active state. A breakthrough application of antibody fragments has been in the field of GPCRs which are highly flexible and sample multiple conformations [156]. Cytochrome *c* oxidase with scFv was the first demonstrated co-crystallization of a membrane protein [157], since then several membrane proteins have been crystallized with antibody fragments scFv and Fab [147,149,158,159,160,161,162]. β_2_AR and A_2A_R are two initial examples of successful application of mouse monoclonal Fabs. Fabs, that recognized a three-dimensional epitope, were generated from monoclonal antibodies raised by injecting purified proteins reconstituted in lipid vesicles into mouse. Fab5 was identified for β_2_AR and Fab2838 for A_2A_R. Fab5 recognized a flexible intracellular loop and stabilized the protein to yield a carazolol-bound β_2_AR-Fab structure (PDB 2R4R) [163]. Fab2838, exhibited a broader binding interface that involved interactions with various intracellular loops and cytoplasmic ends of multiple TM helices. It stabilized an inactive conformation that blocked agonist occupancy of the A_2A_R (PDB 3VG9) [164] (Figure 7C(i)). Nanobodies (Nbs) are more advantageous than other antibody fragments as discussed above, and hence have taken precedence over other antibody fragments in multiple in vitro and in cellulo applications. Fabs and T4 lysozyme helped solve the structure of β_2_AR bound to inverse agonist carazolol but failed to produce crystals of good quality in the presence of agonist. Nb80 produced from llama immunized with purified agonist bound β_2_AR in phospholipid vesicles showed G-protein-like properties upon binding to β_2_AR. Nb80- β_2_AR structure showed an outward displacement of TM5/6 and inward movement of TM7/3 on the cytoplasmic face, relative to the inactive structure. The CDR3 region of the Nb inserted itself into the cytoplasmic crevice of the receptor stabilizing the active conformation (PDB 3P0G) (Figure 7C(ii)) [165]. Though Nb80 gave the structure of β_2_AR with high-affinity ligand, it failed to accommodate natural ligands with low-binding affinity highlighting the importance of subtle conformational rearrangements accommodated by the receptor. A clever workaround was the use of directed evolution which led to the identification of Nb6B9. Remarkably, this nanobody showed increased ability to bind the low-affinity agonist providing structures of β_2_AR in the presence of agonists with high to low affinity (PDB 4LDE) [166]. The largest structural change between the active and inactive receptor is the cytosolic movement of TM6 towards the extracellular face breaking the ionic lock. Nb60, selective towards the inactive conformer was shown to stabilize the ionic lock (PDB 5JQH) [167]. A partial agonist bound β_2_AR structure stabilized by Nb71 showed an intermediate movement in the cytosolic face of TM6 (Figure 7C(ii)) providing an explanation for weaker G protein efficacy and possible ligand bias (PDB 6MXT) [168]. The assortment of nanobodies altogether provided various snapshots of an otherwise highly flexible ensemble of conformational states of β_2_AR. In trying to circumvent the immunization of llamas, synthetic nanobodies have more recently been employed to arrest conformational dynamics in flexible membrane proteins [169,170]. 

Lactose permease from *E. coli.* (LacY) catalyzes the coupled transport of lactose and proton across the membrane. It forms an N- and C-terminal six-helix bundle. HDX-MS experiments showed that LacY was highly dynamic for virtually all backbone amide protons [171]. Initial structures of LacY were obtained in inward-open conformation. To resolve the more challenging outward-open state, bulky residues were introduced into tight Gly–Gly interdomain to prevent periplasmic closure [172]. In a study by Smirnova et al. thirty-two nanobodies were generated against the outward-open mutant with the intention of favoring unobserved periplasmic open conformations. Of them, nine exclusively stabilized the outward-open conformation with increased on-rates for substrate binding. Further fluorescence quenching experiments demonstrated that different Nbs stabilized potentially different outward-open states [173]. Nb9039, and Nb9047 were co-crystallized with LacY in its apo form (PDB 5GXB) or bound to 4-nitrophenyl-α-D-galactoside (NPG) (PDB 6C9W), respectively. In the apo state, Nb9039 interacted with major residues in the C-terminal half of LacY while the CDR3 region from the Nb was inserted into the crevice of the N- and C-terminal halves of LacY, thus stabilizing the conformation. The structure of LacY with Nb9047 complexed to substrate NPG was obtained following the apo structure. This structure showed a more occluded vestibule, captured by the substrate NPG, than the apo state with major differences clustered around the periplasmic helices in TM1-6 [174] (Figure 7C(iii)). Interestingly, the periplasmic cavity was found to be more open in the apo form than the substrate-bound form. Thus, subtle differences in conformational flexibility within the same outward-open state were observed using nanobodies. 

In addition to structure determination of trapped conformers, nanobodies can also be utilized to probe the activation of receptors in mammalian cells. Canonical signaling by GPCR is mediated by coupling to heterotrimeric G proteins in the plasma membrane. Evidence supporting the activity of GPCR in endosomes was provided by in cellulo use of conformation specific nanobodies. Activated β_2_AR undergoes phosphorylation and recruits β-arrestins, blocking G protein association, and promoting endocytosis via clathrin-coated pits (CCPs). Nb80 selectively binds the agonist-bound β_2_AR and stabilizes the active receptor conformation. When expressed at a low concentration, Nb80 acts as a sensor for receptor activation. In one study, after the application of the agonist isoprenaline, cytosolic GFP-Nb80 was recruited to the plasma membrane and co-localized with β_2_AR. After some time, upon receptor internalization, GFP-Nb80 disengaged from the receptor and was re-recruited at later timepoints. In this second phase, GFP-Nb80 gets recruited to internalized β_2_AR on endosomes observed as highly mobile puncta. Similar results were obtained from using Nb37 that recognizes the guanine-nucleotide-free form of G_αs_ representing the catalytic intermediate of G protein activation. This demonstrated that endosomal β_2_AR was catalytically active. This observation was later corroborated by using cAMP, a classical second messenger carrying the downstream signal. Overall, in cellulo experiments using Nb80, a conformational biosensor, revealed a discrete signaling response emanating from endosomes [175]. 

Epidermal growth factor receptor (EGFR) is a single transmembrane protein with an extracellular domain that binds substrate, and an intracellular kinase domain that upon phosphorylation relays signal transduction. There are four members in the family which upon ligand binding, such as EGF, form homo or heterodimers upon activation. In one study Nevoltris et al. three nanobodies were generated against EGFR, of which G10 Nb recognized the elongated active receptor while Nbs D10 and E10 recognized different epitopes on the inactive receptor. D10 and E10 were bound to sites other than the active site and behaved as negative allosteric modulators by stabilizing the inactive form. Strong FRET signals from the heterodimeric EGFR receptors were observed while using all three Nbs in the absence of EGF. The addition of the substrate abrogated signals from D10 and E10 Nbs but showed higher intensity for G10. This proved that D10 and E10 recognized epitopes on inactive receptors while G10 recognized both states but had a higher affinity for the active conformer. The formation of heterodimers in the absence of a ligand demonstrated the presence of pre-dimers, where EGFR adopts an inactive state. The pre-dimers eventually switch to the active extended conformers in the presence of ligand. Overall, the three nanobodies acted as a conformational sensor that helped identify a pre-dimer state prior to activation by substrate [176]. 

## 9. Computational Approaches 

Computational methods in general are deeply embedded into the universal fabric of modern science. In biochemistry and biophysics, these methods are increasingly sophisticated and essential for integrating data acquired from disparate technical approaches and complex experimental designs. All high-resolution structural techniques (X-ray, NMR, cryo-EM) strongly depend on computation to produce structure models compatible with acquired data (diffraction patterns, distance/angles restraints, 2/3D class averages, etc.). And since many IMPs and their complexes, particularly involved in transport and signal transduction, are not rigid in nature, computational strategies are needed that are inclusive of these endogenous flexibilities. Assessing the interplay between conformational diversity and dynamics becomes necessary to define their functionality. Static structural characterization provides structural snapshots that seldom reflect functional activity. To better understand the mechanism of action, experimental data on structural transitions, conformational fluctuations, binding and dissociation, etc., must be incorporated into model building. We envision a future where we can download data from repositories such as PDB with movies of biomolecular machines in motion. 

A combination of pure computational approaches with real data analysis to generate dynamic structural ensembles, best representing experimental outcomes, can be achieved in several ways. A recent review by Cardenas et al. [177] provides concise guidance for non-computer specialists to maneuver through this complex endeavor. Briefly, three major strategies are presented—independent approach, guided simulation, and search and select (reweighting) approach. 

The first step in a computational molecular simulation usually consists of sampling different conformations. It can be performed using a detailed atomistic or coarse-grained (less detailed) representation. Sampling protocols such as molecular dynamics (MD), Monte Carlo simulation (MC), etc. vary significantly [178]. In the independent approach, experimental and computational protocols are performed independently, and the results of both methods are compared. In a guided simulation, experimental data is used to constrain the three-dimensional conformation sampling in the computational protocol, usually implemented by the addition of external energy terms. This type of guided simulation has been used in many popular programs such as CHARMM [179], GROMACS [180], Xplor-NIH [181], ENSEMBLE [182] and Phaistos [183]. In the search and select approach, the computation method is performed first to generate a large ensemble of different conformations, and then experimental data is used to choose the correlated ensemble’s population. Different protocols, based on maximum entropy or maximum parsimony, may be used to select conformations that best fit the data [184,185].

To date, most of the application has been directed towards soluble proteins with an uptick in the expansion of these applications to IMPs. For example, cryo-EM density-guided iterative Rosetta-MD protocol, previously developed for soluble proteins, was extended to membrane proteins and tested with a set of five benchmark IMP structures, generated by Rosetta-MD, with initial RMSDs (root-mean-square deviations) of ~3–5 Å from the native conformation [186]. The proposed combined protocol was able to refine these structures to an average RMSD of 1.66 Å by incorporating restraints from 4 Å resolution cryo-EM density maps. Another interesting study combining structural and computational approaches revealed the mechanism of ATP permeation through the voltage-dependent anion channel (VDAC) which mediates metabolite and ion flow across the outer mitochondrial membrane of all eukaryotic cells [187]. The high-resolution structure of VDAC showing a 19-stranded β-barrel with an α-helix partially occupying the central pore was used in MD simulations to construct a Markov State Model of ATP permeation. These simulations indicated that ATP flows through VDAC using multiple pathways, consistent with experimentally determined physiological rates. Additionally, it is important to note that different structural models can result from different experimental approaches—influenza A M2 proton channel serves as one such example. Zhou and Cross have emphasized the importance of membrane mimetic environments and integration with computation (QM/MM) to provide a detailed understanding of the functional mechanism for proton selectivity, conductance, and gating of this important drug target [188].

In addition, guided docking is a different category of the computational method that predicts the final structure of a complex starting from two free molecules. Docking programs, such as HADDOCK [189], IDOCK [190] and pyDockSAXS [191], allows for the incorporation of experimental restraints. For HADDOCK, Koukos et al. have generated the first membrane protein-protein docking benchmark composed of 37 targets with diverse folds and functionality [192]. With this analysis, HADDOCK showed promising docking performance for membrane systems, though it required further optimization. The resulting set of docking decoys, together with analysis scripts, was made freely available for the optimization of membrane complex-specific scoring functions. Another soluble protein-protein docking software, JabberDock, has been adopted for membrane proteins. JabberDock’s defining feature is its usage of a novel protein volumetric representation called spatial and temporal influence density (STID) maps, which are built from short molecular dynamics simulations. Rudden and Deglacomi have presented a pipeline protocol to use this approach for cases involving IMP dimers [193]. They verified JabberDock’s ability to yield accurate predictions in a benchmark of 20 transmembrane dimers, returning a success rate of 75.0%.

Finally, one cannot conclude the progress in computational approaches without mentioning the recent excellent progress in modern AI-based methods. Alphafold2 by DeepMind, for example, provides a database of predicted structures to the scientific community with plans for a wider expansion, to cover more proteins by the end of 2022. By utilizing novel neural network architectures and training procedures based on the evolutionary, physical and geometric constraints of protein structures [194], Alphafold2 remarkably predicts conformations of domains with well-defined (found in exiting structural databases) folds. However, as for now, the program is much less reliable with respect to flexible regions or inter-domain orientations, which are essential for understanding signal transduction or receptor activation.

## 10. Conclusions

In this review, we present a multitude of techniques through various examples, that highlight the progress made towards understanding confirmational dynamics and intrinsic flexibility in membrane proteins. With the aim of studying IMPs in native cellular environments, there is a constant push to develop methodologies that align with this goal. Significant technological developments, thus far, in various structural biology techniques have enabled us to visualize several important members of this diverse collection of proteins. Ever-increasing instrument sensitivities and more sophisticated algorithms grant experimental access to the full-time scale of protein motions and enable a paradigm built on dynamism rather than artificial stasis. Our discourse on the application of various techniques towards understanding conformational flexibilities should prove helpful in envisioning a cross-platform application that could potentially solve or address questions regarding movements within a membrane protein. An obvious example is GPCRs, discussed in several subsections, that underscore the use of multifarious techniques to delineate subtle structural variabilities arising from binding to a ligand or drug. The information gleaned from such an investigation could eventually prove crucial in enhancing drug specificity. Given the breakneck pace of innovation, one can easily envision a not-so-distant future, where solving a structure and extracting information on dynamics are contemporaneous. Moreover, the catalog of sensitive probes continues to swell and further bolsters the prospects of such investigations. Overall, we believe that the broad spectrum of techniques covered herein should supplement the reader with experimental options available in their quest towards correlating confirmational changes in membrane proteins with their downstream signaling events.

## Figures and Tables

**Figure 2 membranes-12-00227-f002:**
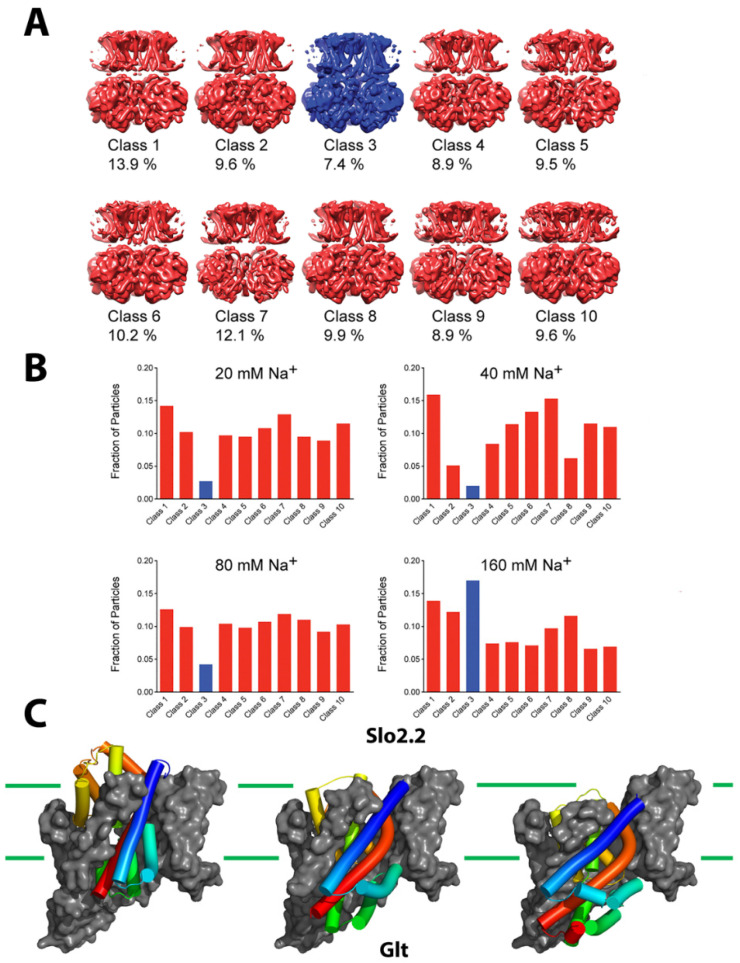
(**A**) 3D class averages from a combined dataset of Slo2.2 vitrified in the presence of 20 mM, 40 mM, 80 mM, and 160 mM Na^+^ concentrations. Classes that resemble closed Slo2.2 are colored red and open colored blue. (**B**) A fraction of particles in each class is shown for the 20–160 mM Na^+^ concentrations. Reproduced with permission from Elsevier. (**C**) Structures illustrate a twisting elevator movement to effect substrate traversal across the bilayer. Transport domain is depicted as colored cylinders and scaffold domain as dark grey surface. For clarity, only one protomer of the obligate homotrimer is shown. Models were resolved by X-ray crystallography, cryo-EM, and aided by engineered disulfide trapping. Left panel: outward-open Glt_Ph_ (PDB 2NWW); middle panel: intermediate, putative Cl^−^ conducting state of Glt_Ph_ (PDB 6WYK); right panel: inward-open (TBOA-inhibited) state of Glt_Tk_ (PDB 6XWR).

**Figure 3 membranes-12-00227-f003:**
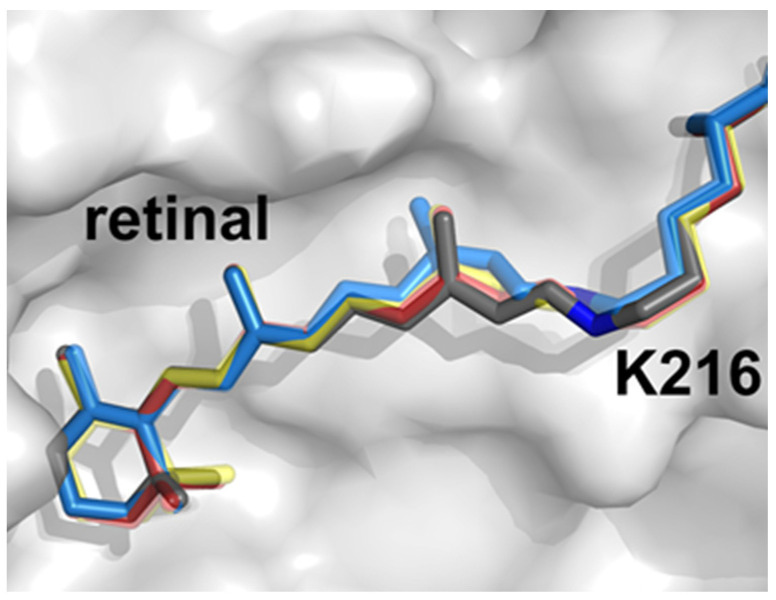
bR retinal *trans*-to-*cis* isomerization observed by ultrafast TR-SFX. Dark grey indicates the dark state, red and orange intermediates (49–406 and 457–646 fs), and blue the 13-*cis* isomer (10 ps) prior to relaxation. Illustration derived from PDB 6G7H, 6G7I, 6G7J, 6G7K.

**Figure 4 membranes-12-00227-f004:**
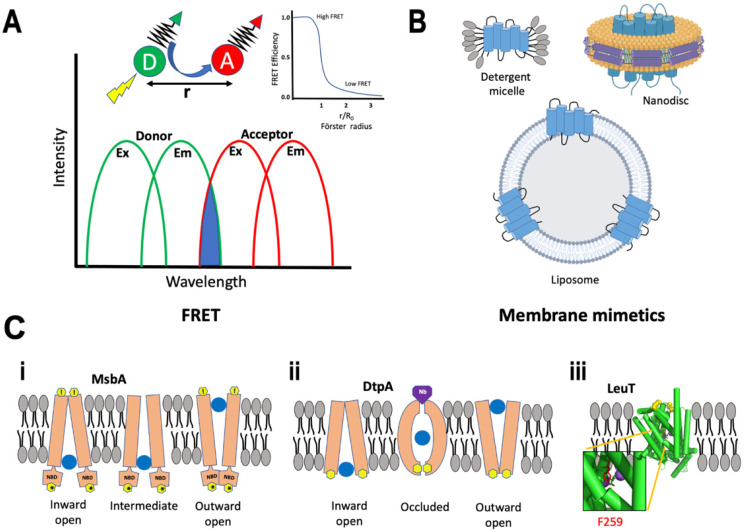
(**A**) Schematic showing the overlap of the donor emission and acceptor excitation spectra leading to the observation of FRET. Additionally shown in inset is the effect of R_0_ on FRET intensity. (**B**) Different membrane mimicking systems available for in vitro reconstitution of membrane protein for FRET studies. Nanodisc image was generated using BioRender. (**C**) The different conformational intermediates occupied by MsbA (i) and DtpA (ii) during the transport of substrate molecule. Nanobody against the closed periplasmic conformer of DtpA is shown in purple. The position of F259 residue in TM helix 6b of LeuT (PDB 3GJD) is shown that acts as a volumetric sensor (iii).

**Figure 5 membranes-12-00227-f005:**
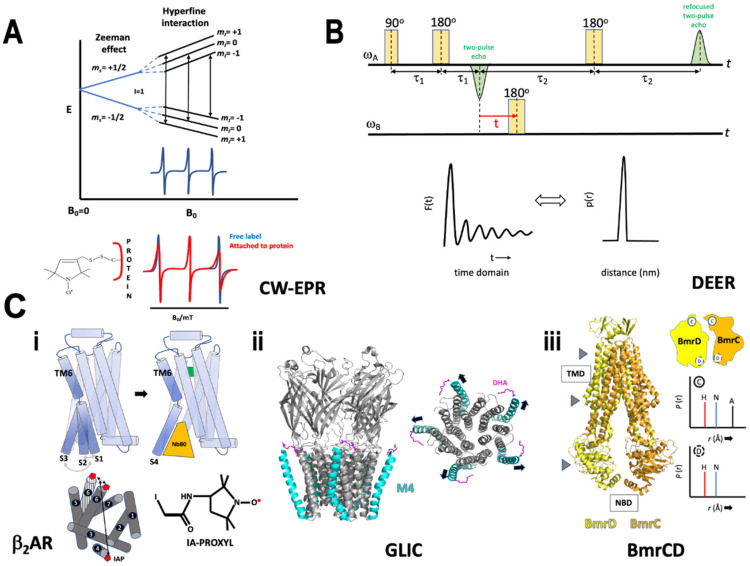
(**A**) Energy level diagram of MTSL. Interaction of the nuclear and electron spins causes further splitting of the spin system with different energy levels known as hyperfine interaction. EPR spectrum of the free spin label (blue) overlaid onto the spectrum of spin label conjugated to a protein (red), bottom panel. (**B**) The DEER four-pulse sequence. For interacting spins, the modulation observed from the refocused echo in the time domain is directly used to measure distances using model-independent analysis. (**C**) (i) The various states of TM6 in β_2_AR. Inactive states (S1, S2); active intermediate state S3 in the presence of agonist (green); and active state S4 with Nb80. Positions of spin-label, IA-PROXYL on TM4 and TM6 for EPR measurements are shown in red. (ii) Structure of GLIC (PDB 5J0Z) highlighting the outermost helix M4 (blue) that interacts with DHA (magenta). Top view without the extracellular domain showing the outward movement of M4. (iii) Structure of BmrCD (PDB 7M33) with protomers BrmD (in yellow) and BmrC (in orange). Arrows indicate position of spin labels. Schematic showing the consensus (C, solid circle), and degenerate NBS (D, broken circle) along with a schematic of DEER distance measurements indicating the presence of HES (H), nucleotide bound (N) and apo (A) conformational states for each of the NBSs.

**Figure 6 membranes-12-00227-f006:**
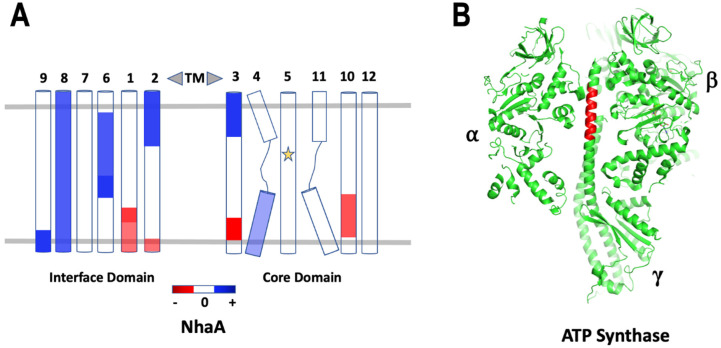
(**A**) Schematic showing the extent of deuterium exchange using a color gradient from red (decreased)—white (unchanged)—blue (increased) in the TMs of NhaA. The yellow star denotes the binding site for Li^+^. (**B**) The structure of F_0_F_1_-ATP synthase (PDB 3OAA) highlights the C-terminal helix in the γ subunit which shows maximum exchange.

**Figure 7 membranes-12-00227-f007:**
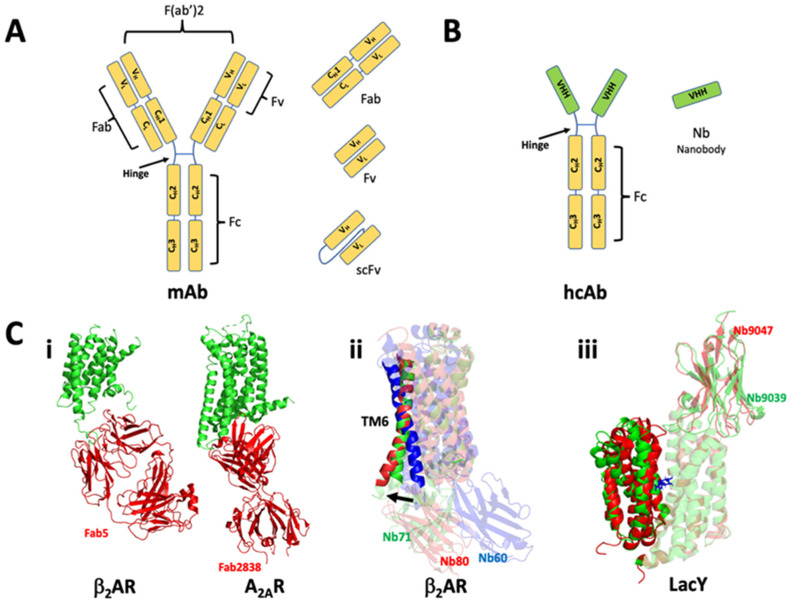
(**A**) Conventional monoclonal antibody (mAb). Various fragments obtain from chemical or enzymatic cleavage. (**B**) Heavy chain antibody (hcAb) found in camelids. The small (~15 kDa) variable region (VHH) from hcAb is called a nanobody. (**C**) (i) Ab fragment stabilized conformers. Structure of β_2_AR and A_2A_R stabilized by antibody fragments Fab5 (PDB 2R4R) and Fab2838 (PDB 3VG9). (ii) Alignment of β_2_AR structures in the presence of different conformational Nbs 60 (5JQH), 71 (6MXT), 80 (3P0G), mimicking a transition from inactive to active state, indicated the outward movement of intracellular TM6. (iii) Overlay of outward-open structures of LacY in the apo and substrate NPG (blue) bound state, conformationally trapped by Nb 9039 (5GXB) and 9047 (6C9W) respectively.

## Data Availability

Not applicable.

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
