# Peer review of "New Horizons in Structural Biology of Membrane Proteins: Experimental Evaluation of the Role of Conformational Dynamics and Intrinsic Flexibility"

_membranes, 2022, doi:10.3390/membranes12020227_

Round 1
Reviewer 1 Report
This is excellent and timely review on a series of structural biology techniques to study membrane protein structure/dynamics. In each technique, the authors provided brief theoretical background followed by multiple examples/publications, which gives the readers very good combination for understanding how each technique works and is applied to the membrane protein studies. I only have several minor suggestions:
1. Excellent examples have been given to use 19F NMR to study multiple conformational states of membrane proteins such as GPCR. Would the authors comment briefly about the utility of 2D-based 15F NMR experiments such as 19F-13C TROSY?
2. For CryoEM section, perhaps mention the current limitation to study flexible regions of the membrane proteins such as single pass transmembrane proteins, for example, Schumacher et al. Sci Adv. 2021 May 7;7(19):eabe9716 were not able to observe the transmembrane-cytoplasmic domain of integrin a5b1 in their cryoEM structural analysis.
3. In FRET analysis, is there a limitation that FRET can best detect intermolecular or intramolecular interaction at distances <100A? Should this be stated as advantage vs other methods?
4. In computational section, would it be useful to briefly mention the AI-based methods such as Alphafold2, which promise to tackle complex membrane protein structures in near future. In fact, there are already some examples in literatures.
Reviewer 2 Report
see attached file
